# Compressive Behavior Characteristics of High-Performance Slurry-Infiltrated Fiber-Reinforced Cementitious Composites (SIFRCCs) under Uniaxial Compressive Stress

**DOI:** 10.3390/ma13010159

**Published:** 2020-01-01

**Authors:** Seungwon Kim, Seungyeon Han, Cheolwoo Park, Kyong-Ku Yun

**Affiliations:** 1Department of Civil Engineering, Kangwon National University, 346 Jungang-ro, Samcheok 25913, Korea; inncoms@kangwon.ac.kr; 2KIIT (Kangwon Institute of Inclusive Technology), Kangwon National University, 1 Gangwondaegil, Chuncheon 24341, Korea; syhan8704@kangwon.ac.kr

**Keywords:** slurry-infiltrated fiber-reinforced cementitious composite, high-performance fiber-reinforced cementitious composite, fiber volume fraction, compressive stress, stress-strain relationship, filling slurry matrix

## Abstract

The compressive stress of concrete is used as a design variable for reinforced concrete structures in design standards. However, as the performance-based design is being used with increasing varieties and strengths of concrete and reinforcement bars, mechanical properties other than the compressive stress of concrete are sometimes used as major design variables. In particular, the evaluation of the mechanical properties of concrete is crucial when using fiber-reinforced concrete. Studies of high volume fractions in established compressive behavior prediction equations are insufficient compared to studies of conventional fiber-reinforced concrete. Furthermore, existing prediction equations for the mechanical properties of high-performance fiber-reinforced cementitious composite and high-strength concrete have limitations in terms of the strength and characteristics of contained fibers (diameter, length, volume fraction) even though the stress-strain relationship is determined by these factors. Therefore, this study developed a high-performance slurry-infiltrated fiber-reinforced cementitious composite that could prevent the fiber ball phenomenon, a disadvantage of conventional fiber-reinforced concrete, and maximize the fiber volume fraction. Then, the behavior characteristics under compressive stress were analyzed for fiber volume fractions of 4%, 5%, and 6%.

## 1. Introduction

Conventional concrete, which is commonly used as a construction material, does not have an adequate resistance level in terms of collisions or explosive loads; thus, concerns are arising about human casualties from the brittle fracturing when reinforced concrete (RC) structures explode [1,2]. With the increasing threat of global terrorism, as well as safety concerns, there is a need for reinforced concrete structures that can withstand the sudden occurrence of dynamic loads like terrorist impact and blast. The energy absorbing capacity of the material plays an important role in developing protective structures. Fiber-reinforced concrete is being popular due to its greater impact resistance properties [3,4]. The possibility of high-performance fiber-reinforced cementitious composites (HPFRCCs) satisfying blast resistance design requirements for external blasts more effectively than conventionally reinforced concrete. The HPFRCCs have been expected to improve such drawbacks of concrete and improve the impact resistance [5]. From previous research, it has been found that the mechanical properties of the concrete can be improved by adding fibers into the mix. Otter and Naaman reported that the addition of fiber in the concrete had a significant effect on the strength and toughness of concrete [6]. The possibility of HPFRCCs satisfying blast resistance design requirements for external blasts more effectively than conventionally reinforced concrete. HPFRCC has been found to be an economical solution against the blast loadings [7].

As the compressive stress of concrete increases, the elastic area becomes larger, and the load-bearing capacity decreases sharply after the manifestation of the maximum strength [8]. The strengthening mechanism of fiber involves the transfer of stress from the matrix to the fiber by interlocking the fibers and matrices when the fiber surface is deformed. The stress is thus shared by the fibers and matrix in tension until the matrix cracks, and then the total stress is progressively transferred to the fibers [9]. These characteristics of high-strength concrete are determined from the size of strain and the shape and rising and falling curves at the manifestation of the maximum strength; these characteristics are also determined by analyzing compressive stress test results of concrete [8]. We examined established prediction equations and found that studies of high volume fractions remained insufficient compared to studies of conventional fiber-reinforced concrete. Furthermore, the mechanical properties of HPFRCCs and high-strength concrete revealed limitations in each prediction equation for the strength and characteristics of contained fibers (diameter, length, volume fraction) even though the stress-strain relationship was determined by these factors [10,11,12,13,14,15,16,17]. Slurry infiltrated fiber concrete (SIFCON) is one of the HPFRCCs. SIFCON is made by distributing short discrete fibers in the mold to the full volume or designed volume fraction and then infiltrated by a fine liquid cement-based slurry or mortar. The fibers can be sprinkled by hand or by using fiber-dispending units for large sections. [15]. Therefore, this study aimed to develop a high-performance slurry-infiltrated fiber-reinforced cementitious composite (SIFRCCs) that could prevent the fiber balling, a disadvantage of conventional fiber-reinforced concrete, and maximize the fiber volume fraction. Most fiber balling occurred during the fiber addition process due to excessive fibers in the mixing of HPFRCCs or fiber-reinforced concrete. Further, the compressive behavior of the developed SIFRCCs was analyzed for fiber volume fractions of 4%, 5%, and 6%. Cylindrical specimens were fabricated for each variable according to the mixing design of high-performance filling slurry, and an experimental study was conducted on the behavior characteristics under compressive stress to examine the mechanical properties of SIFRCCs with a high volume fraction. 

## 2. Mechanical Properties of Fiber-Reinforced Concrete

Previously proposed stress-strain relationship equations have been modified based on the research results of Propovics [10] and Sargin [11]. Propovics [10] defined the stress-strain relationship through the rate of decrease of the elastic modulus, as shown in Equation (1), and Sargin [11] reflected the properties of a section through the changes of the components and the strength of concrete, as shown in Equation (2).
(1)$$\frac{f_{c}}{f_{ck}}=\frac{\beta \left(\varepsilon /\varepsilon_{0}\right)}{\beta -1+{\left(\varepsilon /\varepsilon_{0}\right)}^{\beta }}$$
(2)$$\frac{f_{c}}{f_{ck}}=\frac{A\left(\varepsilon /\varepsilon_{0}\right)+B{\left(\varepsilon /\varepsilon_{0}\right)}^{2}}{1+C\left(\varepsilon /\varepsilon_{0}\right)+D{\left(\varepsilon /\varepsilon_{0}\right)}^{2}}$$

Here, $f_{c}$ is the stress acting on the concrete; $f_{ck}$, the compressive stress of concrete; ε, the compressive strain; $\varepsilon_{0}$, the strain at peak stress; and β, a coefficient that determines the slope and shape of the curve. Furthermore, A~D are coefficients determined by the boundary condition.

A review of existing studies’ results revealed that the mechanical properties of concrete, including the maximum strength, elastic modulus, and strain at maximum strength, need to be defined before the stress-strain relationship of concrete under compressive stress can be defined. Of these, the elastic modulus is an important design variable for a concrete structure and has a significant effect on the stress-strain relationship. The elastic modulus tends to increase with the strength of the matrix and is proportional to the square root or cube root of the compressive stress of the matrix [1]. The strain at peak stress is usually fixed at 0.002 or 0.0022 for conventional concrete (average strength) [8]. However, the measured compressive stress of high-strength concrete has been reported to have a value exceeding 0.002, and many studies have proposed prediction equations of the strain at peak stress that reflect this value [8].

Steel fibers are mixed for increasing the tensile strength of concrete; however, they also change the mechanical properties in compressive stress measurements [8]. The mixing of steel fibers, generally, increases the compressive stress and the strain and elastic modulus at peak stress [8].

The strengthening effect of fibers is realized as an algebraic sum or a magnification of the fiber reinforcing effects for the elastic modulus of the matrix alone [8,17]. The prediction equations for the stress-strain relationship that reflect changes in the mechanical properties with the mixing of fibers revealed that the effect of fibers was reflected only for conventional concrete with no fiber reinforcement [8,9,10,11,12,13,14,15,16,17].

Furthermore, a prediction equation for estimating the elastic modulus of concrete is generally proposed on the basis of an empirical formula using regression analysis of experimental data of various ranges. The elastic modulus of conventional concrete can be estimated using the compressive stress, unit weight of concrete, etc. [8,9,10,11,15]. However, HPFRCCs, such as SIFRCCs, contain steel fibers and other materials in conventional concrete, depending on the mixing conditions, and these added materials can greatly affect the estimation of the elastic modulus. For HPFRCCs, such as SIFRCCs, the volume fraction of each composite material can be calculated according to the composite theory [18,19,20,21]. 

We examined established prediction equations and found that studies of high volume fractions remained insufficient compared to studies of conventional fiber-reinforced concrete. Therefore, we conducted an experimental study of the behavior characteristics under compressive stress to examine the mechanical properties of SIFRCCs with a high volume fraction of fibers.

## 3. Experimental

### 3.1. Materials

#### 3.1.1. Cement

This study used type 1 ordinary Portland cement (OPC), the physical and chemical properties of which are listed in Table 1 [21].

#### 3.1.2. Silica Fume

To realize high-performance and high-strength filling slurry, this study used silica fume (Elkem Korea, South Korea), the physical and chemical properties of which are listed in Table 2.

#### 3.1.3. Aggregates

To improve the filling performance of the slurry and reduce material separation, fine aggregates with diameters of 0.5 mm or smaller were used. No coarse aggregates were used.

#### 3.1.4. Admixture

To improve the filling performance of the slurry, a polycarboxylic acid, the high-performance water reducing agent with high dispersion performance, was used. The admixture used in this experiment had high strength and high flow characteristics, as well as excellent unit water quantity reduction property and material separation resistance. Table 3 lists the characteristics of the used high-performance water-reducing agent.

#### 3.1.5. Steel Fibers

Double-hook steel fibers for conventional concrete with a diameter of 0.75 mm, length of 60 mm, and an aspect ratio of 80 were used. These steel fibers had a density of 7.8 g/cm^3^ and a tensile strength of 1200 MPa. Figure 1 shows the shape of the used steel fibers.

### 3.2. SIFRCCs

SIFRCCs that can accommodate a high volume fraction of steel fibers were developed in this study to prevent the fiber ball phenomenon, the main disadvantage of conventional fiber-reinforced concrete, and maximize the fiber volume fraction [22,23,24,25,26]. Unlike conventional fiber-reinforced concrete, SIFRCCs is a type of HPFRCC that can contain a high volume of steel fibers. It is fabricated by filling steel fibers by dispersing them and then filling high-performance slurry. SIFRCCs affords the advantages of preventing the fiber ball phenomenon and allowing a high fiber volume fraction [23,24,26].

### 3.3. Mixing and Fabrication of Specimens

OPC, fine aggregate, water, superplasticizer, and additional silica fume were mixed to prepare a slurry. The water binder ratio was 0.35. First, the cylindrical mold with diameters of 100 mm and heights of 200 mm were filled with steel fiber with respect to volume fraction. Randomly sprinkled steel fibers in the mold should not overfill the depth of mold and level up as much as possible [25,26]. The slurry as prepared after mixing the contents was poured until no more bubbles were seen to ensure infiltration of slurry into the fibers because the void has negative effects on the strength of the concrete. The amount of high-performance water reducing agent was set to 2.5% of the binder weight [25,26]. To reduce the material separation and achieve the required strength, fine aggregates (50% of binder weight) and silica fume (15% of cement weight) were added [25,26]. Table 4 shows SIFRCCs mixing.

To analyze the compressive behavior characteristics of the SIFRCCs for fiber volume fractions of 4% (312 kg per cubic meters), 5% (390 kg per cubic meters), and 6% (468 kg per cubic meters), cylindrical specimens with diameters of 100 mm and heights of 200 mm were fabricated with respect to the mixing ratio of each variable, as listed in Table 4. Figure 2 shows the fabrication of specimens.

### 3.4. Compression Test of SIFRCCs

A universal testing machine (UTM) was used to study the compressive behavior of specimens. The specimens were placed centrally between the two compression plates, such that the center of moving head was vertically above the center of specimen, as shown in Figure 3, then the load was applied on the specimens by moving the movable head. The load and corresponding contraction were measured at different intervals. A review of existing studies on the mechanical properties of high-strength concrete and fiber-reinforced concrete found that there was an applicable strength limit for each prediction equation [8,10,11,12,13,14,15,16,17]. If this strength limit is exceeded, structural design problems can be generated by estimation of the unsafe side. Furthermore, major mechanical properties that define the stress-strain relationship have been found to be determined by the compressive stress of concrete and the fiber volume and shape [8,10,11,12,13,14,15,16,17]. Therefore, this study also used the high fiber volume fraction as a major variable. In addition, an experimental study on the elastic modulus under compressive stress, strain at peak strength, and Poisson’s ratio was performed. Figure 3 shows the experimental setup for compression tests. Compressive strength, modulus of elasticity, and Poisson’s ratio were determined according to ASTM (American Society for Testing and Materials) specifications (ASTM C873 [27] and ASTM C469 [28], respectively).

## 4. Experiment Results and Analysis

### 4.1. Compressive Stress

Figure 4 shows the compressive stress experiment results with respect to the fiber volume fraction for SIFRCCs. For fiber volume fraction of 6%, the average compressive stress was ~83 MPa. For fiber volume fraction of 5%, the average compressive stress was ~75 MPa; this was ~10% lower than that for the fiber volume fraction of 6%. Furthermore, for the fiber volume fraction of 4%, the average compressive stress was ~66 MPa; this was ~12% and ~21% lower than that for fiber volume fractions of 5% and 6%, respectively. The compressive stress tended to increase in proportion to the fiber volume fraction. It seemed that the increased amount of steel fibers with increasing fiber volume fraction produced a restraining effect on the specimen itself, thereby affecting the increase in compressive stress.

Many studies have shown that the mixing of fibers in fiber-reinforced concrete generally does not influence the compressive stress of the matrix itself. The compression experiment results of Shah and Rangan [29] reported that although the compressive strain at fracture increased significantly when fibers were mixed, the compressive stress did not clearly improve. As the fiber volume fraction increased, the compressive stress gradually increased at first; however, above a certain fraction (4% or more), this tendency changed somewhat. Within a certain mixing level of fibers, the compressive stress increased because the resistance was increased by crack suppression; however, above this level, the compressive stress decreased owing to additional defects that appeared with the improvement effect.

The compressive fracture of conventional fiber-reinforced concrete was caused not by the yield or drawing of fibers but by the fracture of a matrix with relatively low strength before fibers play a structural role. Accordingly, the compressive stress was considered to improve with the fiber volume fraction as sufficient adhesion occurred between the fiber and the high-strength cementitious matrix used in this study. Micro-cracks were caused in conventional concrete and fiber-reinforced concrete by the interface properties of the coarse aggregates and the cement paste. For SIFRCCs, a composite made of fine particles with no coarse aggregates, cracks at the interface could be reduced, and the adhesion performance of the filling slurry and steel fibers could be improved. Therefore, the effect of steel fibers, such as the fiber volume fraction, rather than the effect of the compressive stress caused by matrix fracture was considered to be reflected.

### 4.2. Elastic Modulus

The elastic modulus experiment results indicated that the elastic modulus showed insignificant differences with the fiber volume fraction; its value remained ~28 GPa. This was in contrast to the compressive stress experiment results. However, as shown in Figure 5, the strain at peak stress showed significant differences with increasing fiber volume fraction. These results implied that the energy absorption capacity improved as the fiber volume fraction increased.

Figure 6 shows the elastic modulus estimation curve using the compressive stress as presented in KCI 2012 [30] and ACI 318 [31] and the compressive stress and elastic modulus with respect to the fiber volume fraction. The elastic modulus with respect to the fiber volume fraction measured in this study satisfied the lower limit of the elastic modulus estimation curve. This seemed to be because the elastic modulus prediction equation suggested in KCI 2012 [30] and ACI 318 [31] estimated the elastic modulus using only the compressive stress and unit weight.

### 4.3. Poisson’s Ratio

Poisson’s ratio of conventional concrete is known to be 0.16–0.20; that of fiber-reinforced high-strength concrete is expected to be higher. Figure 7 shows Poisson’s ratio with respect to the fiber volume fraction of SIFRCCs. As with the elastic modulus experiment results, Poisson’s ratio showed insignificant differences with the fiber volume fraction. The average Poisson’s ratio was ~0.3; this was much higher than that of conventional concrete and was similar to that of steel. This was because the high-strength filling slurry matrix improved the interface adhesion between the matrix and the steel fibers. SIFRCCs showed homogeneous behavior owing to its high fiber volume fraction; this could suppress brittle fracture, a disadvantage of conventional concrete, and provide sufficient energy absorption capacity.

### 4.4. Stress-Strain Relationship

The stress-strain experiment results of SIFRCCs with respect to the fiber volume fraction showed relatively high strain values for all variables compared to that of conventional concrete, with a strain of 0.008 or higher at peak stress. Figure 8 shows the results for the fiber volume fraction of 6%. In this case, the post-peak behavior exhibited a strain hardening behavior. Even after the elastic section in the stress-strain curve, it showed ductile behavior characteristics like those of metal. This showed that as in the experimental results obtained with high Poisson’s ratio, the specimen exhibited sufficient energy absorption capacity even after the peak stress owing to its high fiber volume fraction. Figure 9 shows the results for the fiber volume fraction of 5%. As in the previous case, the post-peak behavior exhibited a strain hardening behavior in the plastic section of the stress-strain curve. Figure 10 shows the results for the fiber volume fraction of 4%. Unlike in the previous two cases, the post-peak behavior exhibited a strain hardening behavior to some degree in the plastic section of the stress-strain curve and a strain-softening behavior in the end. This was because the low fiber volume fraction of 4% produced only a small amount of fiber reinforcement that decreased the resistance to the expansion force in the vertical direction of the specimen axis. Figure 11 shows the results for the stress-strain curve with respect to fiber volume fraction.

Table 5 lists the experimental results for maximum strength, elastic modulus, and strain at peak stress; these are generally used to define the stress-strain curve of the SIFRCCs. The relative error of compressive strength in the average value of the experimental results for each variable with respect to the fiber volume fraction was lower than 10%. The stress-strain experiment results revealed insignificant differences in the elastic modulus with the fiber reinforcement amount. This suggested that fiber reinforcement had no effect on the elastic modulus owing to the use of the same filling slurry matrix regardless of the fiber volume fraction.

However, the strain at peak stress also increased with the fiber volume fraction. In contrast to the above-mentioned compressive stress experiment results with respect to the fiber volume fraction, the compressive stress for the fiber volume fraction of 6% increased slightly by approximately 1.11 and 1.27 times compared to those for fiber volume fractions of 5% and 4%, respectively. However, the strain at peak stress for the fiber volume fraction of 6% increased significantly by 1.51 and 2.67 times compared to those for fiber volume fractions of 5% and 4%, respectively. The compressive fracture patterns obtained by performing the compressive behavior experiment, shown in Figure 12, indicated that the strain increased further as axial cracks in the specimen were restrained. This characteristic of the strain increase rate was considered to be caused by the restraining effect of the high fiber volume fraction, in which many steel fibers resisted the expansion force generated vertically to the specimen axis and thereby caused longitudinal split cracks. Moreover, the steel fiber reinforcement effect served to increase the size of the peak stress and the strain at peak stress in all compressive stress areas.

## 5. Conclusions

A high-performance SIFRCCs that can prevent the fiber ball phenomenon, a disadvantage of conventional fiber-reinforced concrete, and maximize the fiber volume fraction was developed, and the compressive behavior characteristics with respect to the fiber volume fraction were analyzed. The following conclusions were derived from this study.
(1)The static compressive behavior characteristics with respect to the fiber volume fraction of SIFRCCs showed that the compressive stress increased in proportion to the increasing fiber volume fraction. The micro-cracks in conventional concrete and fiber-reinforced concrete were caused by the interface properties of coarse aggregates and cement paste. By contrast, SIFRCCs could reduce cracks at the interface because it was a composite made of fine particles with no coarse aggregates. Furthermore, the filling slurry and steel fibers improved the adhesion performance and reflected the effects of the fiber volume fraction and steel fibers instead of the compressive stress characteristics caused by matrix fracture. In addition, the high volume fraction of the steel fibers generated a restraining effect in the compressive stress test specimen, thereby affecting the increase in compressive stress.(2)The elastic modulus experiment results showed that the elastic modulus did not increase with the increasing fiber volume fraction, unlike the compressive stress experiment results. However, for strain under compressive stress, the increased rate of the strain at peak stress showed a difference of up to 2.7 times depending on the fiber volume fraction. This means that with increasing fiber volume fraction, the post-peak behavior after peak stress showed a strain hardening behavior, implying that the energy absorption capacity improved at a higher fiber volume fraction.(3)The characteristics of the strain increase rate were attributed to the restraining effect of the high fiber volume fraction in which many steel fibers resisted the expansion force generated vertically to the specimen axis that caused longitudinal split cracks. Furthermore, the reinforcing effect of steel fibers was considered to increase the peak stress and strain at peak stress in all compressive stress areas.(4)Poisson’s ratio experiment results showed insignificant differences in the elastic modulus with the fiber volume fraction. Similarly, Poisson’s ratio showed insignificant differences with the fiber volume fraction. However, Poisson’s ratio was ~0.3 for every variable; this was similar to that of steel. This was caused by the improved interface adhesion performance between the high-strength filling slurry matrix and the steel fibers. Therefore, the SIFRCCs with a high fiber volume fraction could suppress brittle fracture, a disadvantage of conventional concrete, and exhibit sufficient energy absorption capacity owing to its homogeneity.

## Figures and Tables

**Figure 1 materials-13-00159-f001:**
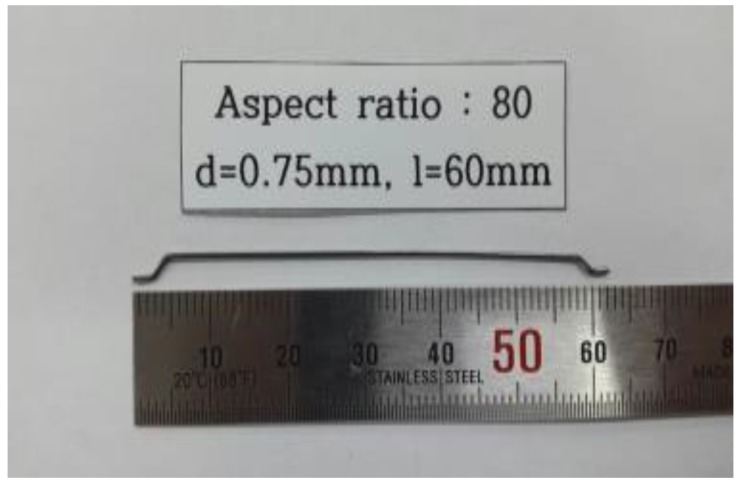
The shape of the used steel fiber.

**Figure 2 materials-13-00159-f002:**
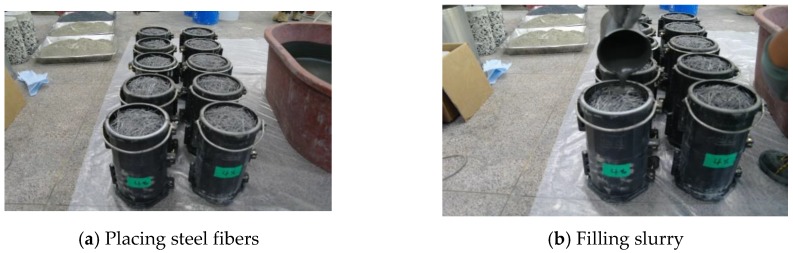
Fabrication of specimens.

**Figure 3 materials-13-00159-f003:**
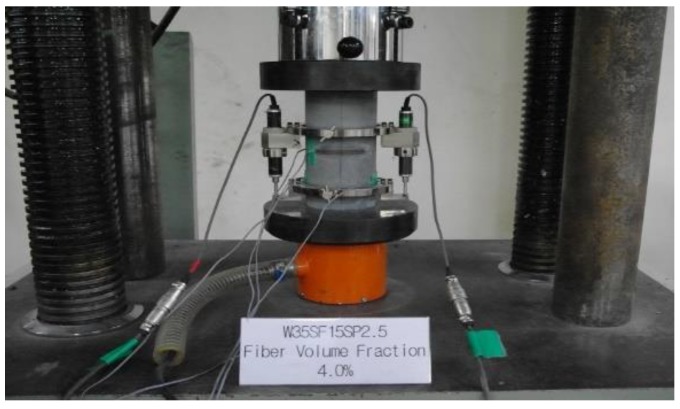
Experimental setup for compressive stress and elastic modulus tests.

**Figure 4 materials-13-00159-f004:**
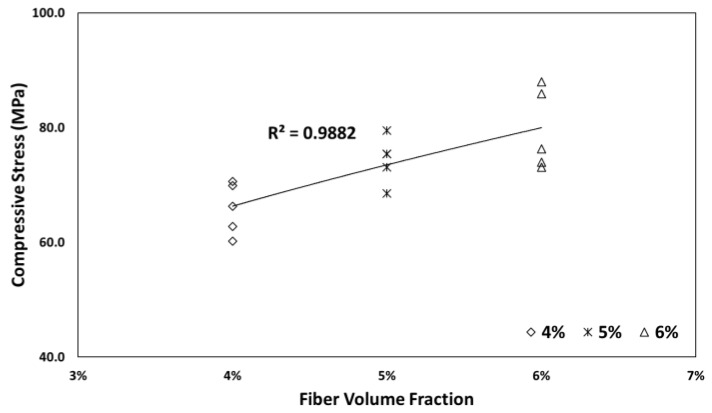
Compressive stress experiment results with respect to fiber volume fraction.

**Figure 5 materials-13-00159-f005:**
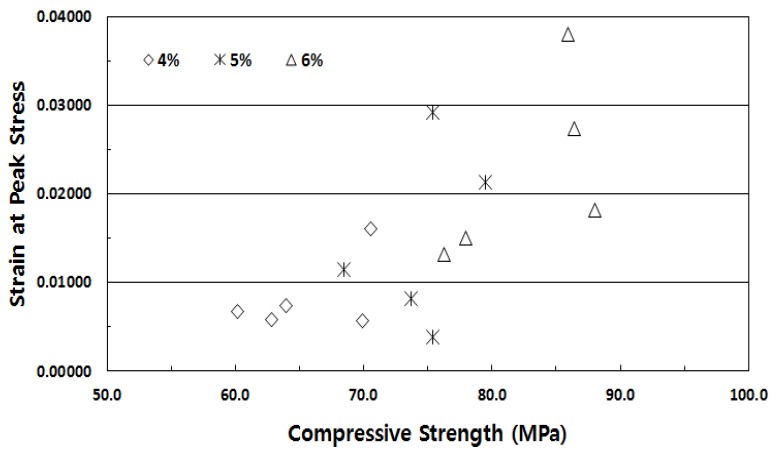
Strain at peak stress.

**Figure 6 materials-13-00159-f006:**
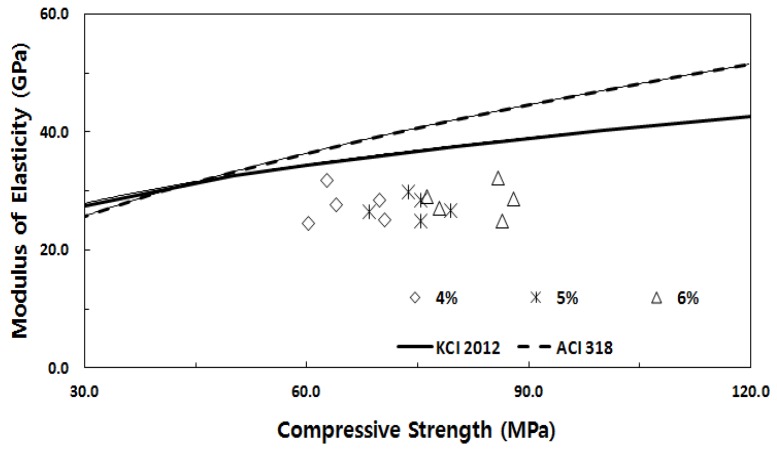
Correlations of compressive stress and elastic modulus between KCI 2012 [30] and ACI 318 [31].

**Figure 7 materials-13-00159-f007:**
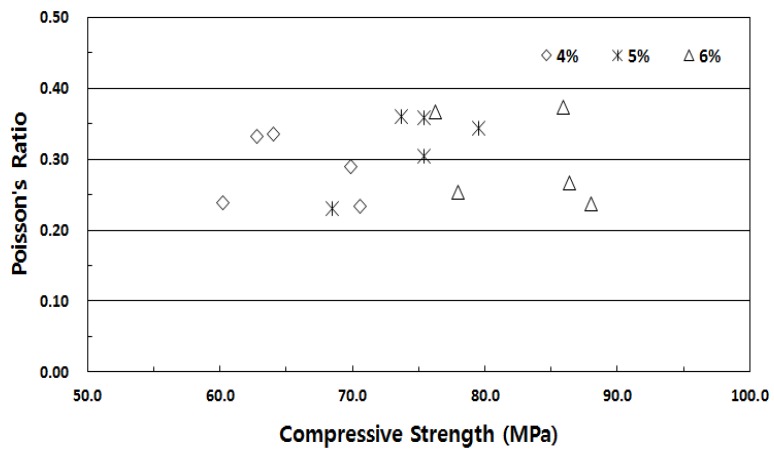
Correlation between compressive stress and Poisson’s ratio with respect to fiber volume fraction.

**Figure 8 materials-13-00159-f008:**
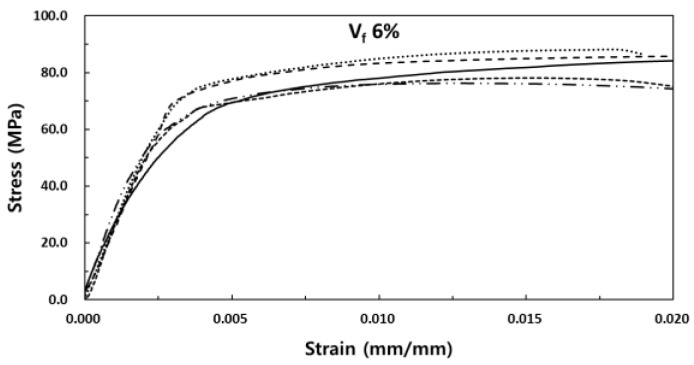
Stress-strain curve for the fiber volume fraction of 6%.

**Figure 9 materials-13-00159-f009:**
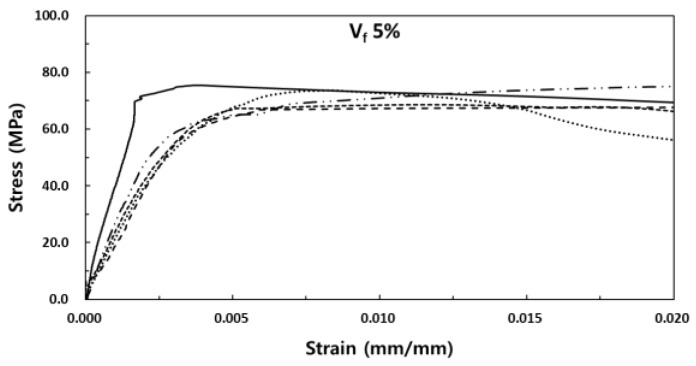
Stress-strain curve for the fiber volume fraction of 5%.

**Figure 10 materials-13-00159-f010:**
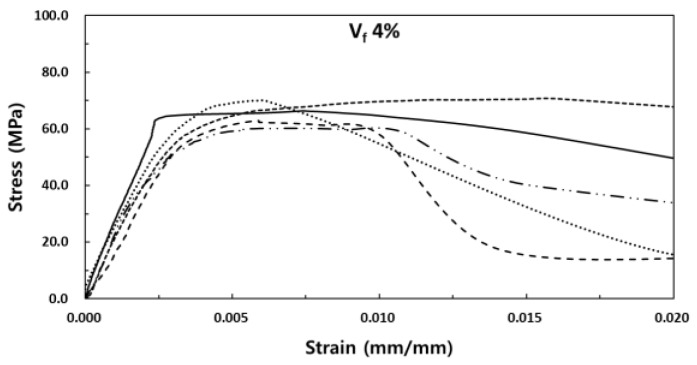
Stress-strain curve for the fiber volume fraction of 4%.

**Figure 11 materials-13-00159-f011:**
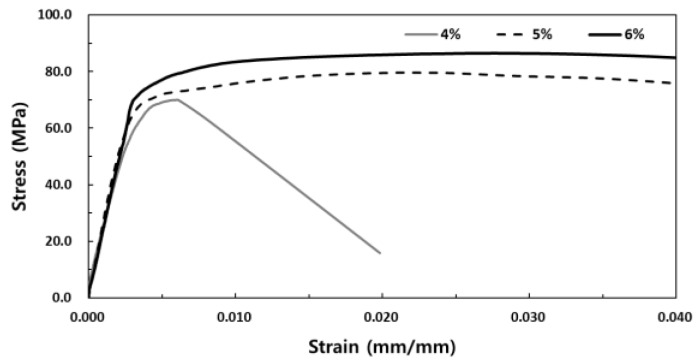
The stress-strain curve with respect to fiber volume fraction.

**Figure 12 materials-13-00159-f012:**
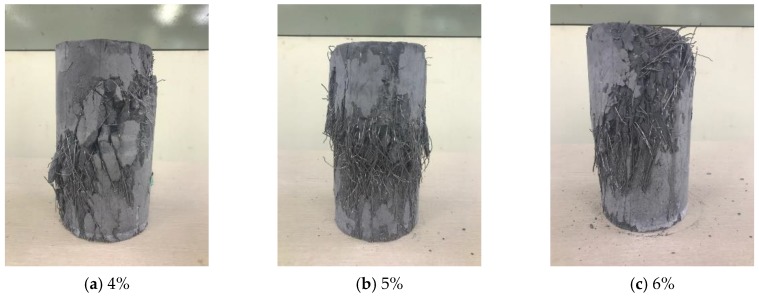
Compressive fracture pattern with respect to fiber volume fraction.

**Table 1 materials-13-00159-t001:** Physical and chemical properties of the used cement.

Physical Properties					
Specific Gravity	Blaine (cm^2^/g)	Stability (%)	Setting Time (min)		Loss on Ignition (%)
			Initial	Final	
3.15	3400	0.10	230	410	2.58
**Chemical Composition (%, mass)**					
SiO_2_	CaO	MgO	SO_3_		Al_2_O_3_
21.95	60.12	3.32	2.11		6.59

**Table 2 materials-13-00159-t002:** Physical and chemical properties of the used silica fume.

**Physical Properties**				
**Specific Gravity**		**Blaine (cm^2^/g)**		
2.10		200,000		
**Chemical Composition (%, mass)**				
SiO_2_	CaO	MgO	SO_3_	Al_2_O_3_
96.00	0.38	0.10	-	0.25

**Table 3 materials-13-00159-t003:** Characteristics of the used high-performance water reducing agent.

Principal Component	Specific Gravity	pH	Alkali Content (%)	Chloride Content (%)
Polycarboxylate	1.05 ± 0.05	5.0 ± 1.5	<0.01	<0.01

**Table 4 materials-13-00159-t004:** SIFRCCs (high-performance slurry-infiltrated fiber-reinforced cementitious composite) mixing proportion.

Fiber (% vol.)	W/B (Water-Binder Ratio)	Unit Material Quantity (kg/m^3^)					
		Water	Cement	Fine Aggregate	Silica Fume	Superplasticizer	Steel Fiber
4%	0.35	407.4	962.8	566.4	169.9	28.3	312
5%							390
6%							468

**Table 5 materials-13-00159-t005:** Experimental results for compressive behavior characteristics with respect to the fiber volume fraction.

Variables ($V_{f}$)	$f_{c}$ (MPa)	$E_{c}$ (MPa)	$\varepsilon_{co}$ (×10^−6^)	ν
4%	69.9	28,430	5.74	0.289
	70.6	25,098	16.08	0.233
	62.8	31,826	5.86	0.333
	64.0	27,667	7.40	0.336
	60.2	24,493	6.67	0.239
**Average**	**65.5**	**27,503**	**8.37**	**0.286**
5%	73.7	29,853	8.16	0.360
	68.5	26,548	11.52	0.231
	79.5	26,617	21.36	0.343
	75.4	24,804	3.84	0.305
	75.4	28,481	29.16	0.358
**Average**	**74.5**	**27,261**	**14.81**	**0.319**
6%	88.0	28,551	18.18	0.237
	78.0	26,967	15.02	0.253
	86.4	24,886	27.38	0.267
	85.9	32,182	38.02	0.373
	76.3	29,071	13.20	0.367
**Average**	**82.9**	**28,331**	**22.36**	**0.300**

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
