# Peer review of "Compressive Behavior Characteristics of High-Performance Slurry-Infiltrated Fiber-Reinforced Cementitious Composites (SIFRCCs) under Uniaxial Compressive Stress"

_materials, 2020, doi:10.3390/ma13010159_

Round 1

Reviewer 1 Report

COMMENTS TO AUTHORS

The subject under study is interesting. The research process is correct and the methodology used is appropriate. Characterization tests of concrete mixtures are common in this type of study.

However, I propose to the Authors some suggestions with the aim of improving the manuscript for possible publication in Materials. Are the following:

1. Introduction

Line 44: Authors should explain the “fiber ball phenomenon”.

It is important to note that these high strength concretes are actually mortars, since the aggregates used are very fine. What is the opinion of the Authors?

3. Experimental

3.1.1 Cement

Line 54: The authors indicate that for the experimental phase “This study used type 1 ordinary Portland cement”. If ASTM standards have been used as a reference, it is necessary to indicate. Also include this reference in the bibliography “ASTM C150: Standard Specification for Portland cement standard and indicate it in the bibliography”.

Table 1

It is necessary to indicate the units of the specific gravity of the cement. The ASTM has designated five types of Portland cement, designated Types I-V. Physically and chemically these cement types differ primarily in their content of Tricalcium Aluminate (C3A), and in their fineness, measured as specific surface. It’s necessary to indicate Surface Area according to Blaine. To complete the chemical composition of the cement, also include the content of Magnesium Oxide (MgO).

3.1.2 Silica fume

Table 2

It is necessary to indicate the units of the specific gravity of the silica fume used. Replace Fineness with Specific Surface

3.1.3 Aggregates

Line 101: In order to complete the information, it is necessary to indicate the nature of the aggregates used (silicic, limestone, etc.) and their grading granulometric curve.

3.3 Mixing and fabrication of specimens

The water/binder ratio is well expressed in the text (0.35), since in the mortar and concrete dosing processes, the relationship between two components, in this case water and cement, is a number, not a percentage. According to this criterion, in Table 4 indicate W/B 0.35. In Table 4 replace "variables" with Fiber (% Volume) The ratio of components dosed is by weight, and then this should be made explicit in the text. Indicate the dimensions of the cylindrical specimens.

3.4 Compression test of SIFRCCs

It is necessary to indicate the standards ASTM that have been used in the tests of Compressive Strength, Modulus of Elasticity and Poisson's Coefficient (Poisson’s Ratio), and include them in the Bibliography

4.1 Compressive stress

Figure 3

Put in the x-axis: fiber volume (%) References (Bibliography)

Authors should adapt the bibliography to the Materials format. There are many wrongs about it. Please, review it and read the Authors Guide.

Special attention should be to the denomination of the scientific journals referenced in the Bibliography.

Authors should include in the Bibliography all the standards used in experimental laboratory tests

Final comment

I congratulate the Authors for the work done and encourage them to continue their researches.

Author Response

Response to Reviewer 1 Comments

Point 1:

Introduction

Line 44: Authors should explain the “fiber ball phenomenon”.

It is important to note that these high strength concretes are actually mortars, since the aggregates used are very fine. What is the opinion of the Authors?

Response 1:

Thank you for your comments.

It is corrected and additional sentence is added.

Coarse aggregates were not used because they are SIFRCCs in the form of placing steel fibers in advance and filling high-performance slurry to increase fiber volume fraction. Also, fine aggregate was used to produce HPFRCCs with high strength and high performance.

Point 2:

Experimental

3.1.1 Cement

Line 54: The authors indicate that for the experimental phase “This study used type 1 ordinary Portland cement”. If ASTM standards have been used as a reference, it is necessary to indicate. Also include this reference in the bibliography “ASTM C150: Standard Specification for Portland cement standard and indicate it in the bibliography”.

Response 2:

Thank you for your comments.

ASTM C150 was citation.

Table 1

It is necessary to indicate the units of the specific gravity of the cement. The ASTM has designated five types of Portland cement, designated Types I-V. Physically and chemically these cement types differ primarily in their content of Tricalcium Aluminate (C3A), and in their fineness, measured as specific surface. It’s necessary to indicate Surface Area according to Blaine. To complete the chemical composition of the cement, also include the content of Magnesium Oxide (MgO).

Response 2:

Thank you for your comments.

It is corrected and fineness is replaced with blaine.

Specific gravity has no unit and the content of MgO is mentioned in Table 1.

3.1.2 Silica fume

Table 2

It is necessary to indicate the units of the specific gravity of the silica fume used. Replace Fineness with Specific Surface

Response 2:

Thank you for your comments.

Specific gravity has no unit.

It is corrected and fineness is replaced with blaine.

3.1.3 Aggregates

Line 101: In order to complete the information, it is necessary to indicate the nature of the aggregates used (silicic, limestone, etc.) and their grading granulometric curve.

Response 2:

Thank you for your comments.

Grading curve has been added.

3.3 Mixing and fabrication of specimens

The water/binder ratio is well expressed in the text (0.35), since in the mortar and concrete dosing processes, the relationship between two components, in this case water and cement, is a number, not a percentage. According to this criterion, in Table 4 indicate W/B 0.35. In Table 4 replace "variables" with Fiber (% Volume) The ratio of components dosed is by weight, and then this should be made explicit in the text. Indicate the dimensions of the cylindrical specimens.

Response 2:

Thank you for your comments.

It has been corrected and dimensions of the cylindrical specimens is mentioned.

3.4 Compression test of SIFRCCs

It is necessary to indicate the standards ASTM that have been used in the tests of Compressive Strength, Modulus of Elasticity and Poisson's Coefficient (Poisson’s Ratio), and include them in the Bibliography

Response 2:

Thank you for your comments.

It has been corrected citation has been added.

Point 3:

4.1 Compressive stress

Figure 3

Put in the x-axis: fiber volume (%) References (Bibliography)

Authors should adapt the bibliography to the Materials format. There are many wrongs about it. Please, review it and read the Authors Guide.

Special attention should be to the denomination of the scientific journals referenced in the Bibliography.

Authors should include in the Bibliography all the standards used in experimental laboratory tests

Response 3:

Thank you for your comments.

It has been corrected and it is experimental data from our lab. experiments.

Point 4:

Final comment

I congratulate the Authors for the work done and encourage them to continue their researches.

Response 4:

Thank you for your comments.

Reviewer 2 Report

L38-42 “…a review of existing studies on the mechanical properties of high-performance fiber-reinforced cementitious composites (HPFRCCs) and high-strength concrete revealed limitations in each prediction equation for the strength and characteristics of contained fibres (diameter, length, volume fraction) even though the stress-strain relationship is determined by these factors [2-9].” 

It would be advisable to describe precisely what statement is given in which research. This position seems to involve different teams and their material and tests could be different from each other.

There are also other researches that describe the behaviour of slurry infiltrated concrete with fibre reinforcement made in the years 1980 and after, f.e.: “Stress-Strain Properties of SIFCON in Uniaxial Compression and Tension” made by Homrich J. R. and Naaman A.E. from August 1988. It would be advisable to extend the description of literature subjected to SIFC testing and to point out and prove the novelty of your work.

L76-79 Same comments as above.

L162 what certain level of fibers? Could you describe previous findings in more detail, what kind of material has been used in a test?

L220 “This was because the low fiber volume friction of 4%..” Should be friction or fraction? Please explain.

L224 - L226 “Relative error in the average value of experimental results for each variable was lower than 10%...” Could you expand and explain the methodology and results of this estimation? Strain differences in the table varies from 5.86 to 16.08 for 4% fraction and from 3.84 to 21.36 for 5% fraction.

General remark: The quality of charts and figure No. 2 should be improved.

Author Response

Response to Reviewer 2 Comments

Point 1:

L38-42 “…a review of existing studies on the mechanical properties of high-performance fiber-reinforced cementitious composites (HPFRCCs) and high-strength concrete revealed limitations in each prediction equation for the strength and characteristics of contained fibres (diameter, length, volume fraction) even though the stress-strain relationship is determined by these factors [2-9].” 

It would be advisable to describe precisely what statement is given in which research. This position seems to involve different teams and their material and tests could be different from each other.

Response 1:

Thank you for your comments.

It has been corrected and taken for references.

Point 2:

There are also other researches that describe the behaviour of slurry infiltrated concrete with fibre reinforcement made in the years 1980 and after, f.e.: “Stress-Strain Properties of SIFCON in Uniaxial Compression and Tension” made by Homrich J. R. and Naaman A.E. from August 1988. It would be advisable to extend the description of literature subjected to SIFC testing and to point out and prove the novelty of your work.

Response 2:

Thank you for your comments.

Thank you for recommending the SIFCON references. I'll refer to the relevant references.

Point 4:

L76-79 Same comments as above.

Response 4:

Thank you for your comments.

It has been corrected and taken for references.

Point 5:

L162 what certain level of fibers? Could you describe previous findings in more detail, what kind of material has been used in a test?

Response 5:

Thank you for your comments.

It has been corrected.

In general, the increasing of steel fibers does not affect the increase in compressive strength. However, SIFRCCs identified a trend of proportional increase in compressive strength with increasing fiber volume fraction, with a constant volume fraction of 4% or more.

Point 6:

L220 “This was because the low fiber volume friction of 4%..” Should be friction or fraction? Please explain.

Response 6:

Thank you for your comments.

It is replaced with fraction.

Point 7:

L224 - L226 “Relative error in the average value of experimental results for each variable was lower than 10%...” Could you expand and explain the methodology and results of this estimation? Strain differences in the table varies from 5.86 to 16.08 for 4% fraction and from 3.84 to 21.36 for 5% fraction.

Response 7:

Thank you for your comments.

I mean relative error is compare with in compressive strength and sentence has been corrected.

Point 8:

General remark: The quality of charts and figure No. 2 should be improved.

Response 8:

Thank you for your comments.

I will submit to the editor before publication for good quality of charts and figures.

Reviewer 3 Report

Introduction is very poor, please address the background of your research since it is not clear. I do not see any benefit of your research without all details of mixing, casting, testing fresh properties and explaining why would you use infiltration at all? What do you want to show compared to the literature? You didn’t compare data from literature with results from your study.

You cited only 9 references for literature, I am sure there much more studies that only reported.

Why would you use such a long fibers, 60 mm? Where is this used in practice? Why we would infiltrated a slurry?

Line 72-74 Please add references for such a statement.

Line 75-76 Please add the reference.

Line 85-86 Please add the reference. Who wrote the Composite Theory?

Table 1 What do you mean by Stability of cement in Table 1? What type of stability?

How is that measured?

Add the unit for specific gravity.

Line 105 Specify the admixture, which company and concentration?

Add particle size distributions for cement, silica fume and aggregates.

What is maximum aggregate size usually used for fiber-reinforced composites?

What is the binder weight?

Figure 3 Why did you plot compressive strength individually, please report average values and corresponding standard deviations.

Specify size of cylinders.

The data presentation is not appropriate for Figures 3, 4, 5, 6.

The deviations of the results in Table 5 within one mix are too large.

How do you think this will be used in the applications?

What were the fresh properties of your mixtures? What was the workability?

Please, add photos from mixing and casting procedures? How did you place fibers vertically?

The amount of fibers in m3 is too high, almost as water volume.

Author Response

Response to Reviewer 3 Comments

Point 1:

Introduction is very poor, please address the background of your research since it is not clear. I do not see any benefit of your research without all details of mixing, casting, testing fresh properties and explaining why would you use infiltration at all? What do you want to show compared to the literature? You didn’t compare data from literature with results from your study.

Response 1:

Thank you for your comments.

SIFRCC has been developed for the purpose of resisting extreme loads such as impact and explosive loads etc..
This paper does not include introduction to dynamic loads because it is about static compressive strength. This is because, if the contents of the dynamic loads are included, the results of the dynamic test as well as the static test are also required.

Point 2:

You cited only 9 references for literature, I am sure there much more studies that only reported.

Response 2:

Thank you for your comments.

There will be more references. However, there are 9 references to this paper. I will look for additional references and refer them.

Point 3:

Why would you use such a long fibers, 60 mm? Where is this used in practice? Why we would infiltrated a slurry?

Response 3:

Thank you for your comments.

As in response 1, 60mm of long-steel fiber is used to resist impact and explosive loads, so that it can resist energy absorption and large deformation.

Only when the volume fraction of steel fibers is high can they resist impact or explosive loads. In order to increase the volume fraction of steel fiber, it is very difficult to increase the fiber volume fractions such as conventional fiber-reinforced concrete. Thus, materials in the form of placing steel fibers in advance and filling slurry were developed as a way to increase the fiber volume fraction.

This is also mentioned in the text.

Point 4:

Line 72-74 Please add references for such a statement.

Response 4:

Thank you for your comments.

It has been citation.

Point 5:

Line 75-76 Please add the reference.

Response 5:

Thank you for your comments.

Reference has been added.

Point 6:

Line 85-86 Please add the reference. Who wrote the Composite Theory?

Response 6:

Thank you for your comments.

References have been added.

Point 7:

Table 1 What do you mean by Stability of cement in Table 1? What type of stability?

How is that measured?

Add the unit for specific gravity.

Response 7:

Thank you for your comments.

Cement stability is a property where cement is hydrated in a stable manner without causing any abnormalities such as volume change. The stability test of cement measures the length change by placing a test specimen made of standard-led cement paste in an autoclave for three hours and then cooling it to 23 ℃ for 15 minutes.

Specific gravity has no unit.

Point 8:

Line 105 Specify the admixture, which company and concentration?

Response 8:

Thank you for your comments.

The admixture is produced by DongNam Company, South Korea and the concentration is pH 5.0 ±2.0.

Point 9:

Add particle size distributions for cement, silica fume and aggregates.

Response 9:

Thank you for your comments.

Particle size is expressed in Blaine and aggregates particle size was mentioned.

Point 10:

What is maximum aggregate size usually used for fiber-reinforced composites?

Response 10:

Thank you for your comments.

Coarse aggregate in fiber-reinforced concrete was used and its size varies depending on the purpose.

Point 11:

What is the binder weight?

Response 11:

Thank you for your comments.

The binder weight means total weight of cement and silica fume.

Point 12:

Figure 3 Why did you plot compressive strength individually, please report average values and corresponding standard deviations.

Response 12:

Thank you for your comments.

We can find average value of compressive strength in table 5.

Point 13:

Specify size of cylinders.

Response 13:

Thank you for your comments.

Size of cylinders has been updated.

Point 14:

The data presentation is not appropriate for Figures 3, 4, 5, 6.

Response 14:

Thank you for your comments.

It has been updated according with data..

Point 15:

The deviations of the results in Table 5 within one mix are too large.

Response 15:

Thank you for your comments.

Due to the high fiber volume fraction, the deviation occurred large. Further experiments on this topic are ongoing.

Point 16:

How do you think this will be used in the applications?

Response 16:

Thank you for your comments.

It can be used for structures requiring special purpose that can be subjected to impact and explosive loads. It is also use for the safety of structures caused by terrorist attacks as the number of terrorist attacks has increased recently.

Point 17:

What were the fresh properties of your mixtures? What was the workability?

Response 17:

Thank you for your comments.

Because it has to be filling between steel fibers, the filling performance of the slurry is important due to its high flowability.

Point 18:

Please, add photos from mixing and casting procedures? How did you place fibers vertically?

Response 18:

Thank you for your comments.

Photos has been added.

Not vertically, not directionally. As shown in Figure 13, the steel fibers appear vertically destroyed because the steel fibers in the loading direction and the vertical direction resist, the photographs show that the fibers have been injected vertically.

Point 19:

The amount of fibers in m3 is too high, almost as water volume.

Response 19:

Thank you for your comments.

Although the actual amount of steel fiber is volume fractions of 4, 5, and 6 %, the density of steel fiber is 7.8g/m3, which is a lot of used for 1m3.

Round 2

Reviewer 2 Report

Dear Sirs

Most of my concerns have been corrected,

congratulations on your work.

Author Response

Response to Reviewer 2 Comments

Point 1:

Most of my concerns have been corrected, congratulations on your work.

Response 1:

Thank you for your comments.

Reviewer 3 Report

Authors did not come up with improved version of their manuscript. Introduction still lacks the background of the conducted research. Experimental techniques are not well explained. Grading curves of all raw materials were not presented. Grading curve for fine aggregates is not correct.

Furthermore, interpretation of results is limited and there are many questions with regard to casting, mixing and test methods which suggest rejection of this paper.

Author Response

Response to Reviewer 3 Comments

Point 1:

Authors did not come up with improved version of their manuscript. Introduction still lacks the background of the conducted research. Experimental techniques are not well explained. Grading curves of all raw materials were not presented. Grading curve for fine aggregates is not correct.

Response 1:

Thank you for your comments.

Introduction has been improved. Experimental part has been edited.

I don’t have grading curves of raw materials. Because physical properties and chemical proportions were received from manufacture company.

Point 2:

Furthermore, interpretation of results is limited and there are many questions with regard to casting, mixing and test methods which suggest rejection of this paper.

Response 2:

Thank you for your comments.

Casting mixing and test methods were edited.

Round 3

Reviewer 3 Report

Figure 1 does not represent grading of fine aggregates. Please, provide correct grading, since you are showing that max aggregate diameter is 1 mm in Figure 1, while stating in the text to be 0.5mm (3.1.3)

The introduction is not imporved, and backround of study is not clear. Please, provide a table with summary of properties of steel fibres (diameter, length, aspect ration, tensile strength, volume used in matrix), sand in matrix (amount, max aggregate diameter), cement (type and amount) which were used up to know in the literature. 

Junior, S., Ribeiro, P. R., Maciel, P. S., Barreto, R. R., Silva Neto, J. T. D., Siqueira Corrêa, E. C., & Bezerra, A. C. D. S. (2019). Thin Slabs Made of High-Performance Steel Fibre-Reinforced Cementitious Composite: Mechanical Behaviour, Statistical Analysis and Microstructural Investigation. Materials, 12(20), 3297. Okeh, C. A., Begg, D. W., Barnett, S. J., & Nanos, N. (2019). Behaviour of hybrid steel fibre reinforced self compacting concrete using innovative hooked-end steel fibres under tensile stress. Construction and Building Materials, 202, 753-761. Trainor, K. J., Foust, B. W., & Landis, E. N. (2012). Measurement of energy dissipation mechanisms in fracture of fiber-reinforced ultrahigh-strength cement-based composites. Journal of engineering mechanics, 139(7), 771-779. Oesch, T., Landis, E., & Kuchma, D. (2018). A methodology for quantifying the impact of casting procedure on anisotropy in fiber-reinforced concrete using X-ray CT. Materials and Structures, 51(3), 73. Trainor, K. J., Flanders, L., & Landis, E. N. (2019)  3d Measurements to Determine Micromechanical Energy Dissipation in Steel Fiber Reinforced Concrete. In VIII International Conference on Fracture Mechanics of Concrete and Concrete Structures (pp. 1-11). Herrmann, H., Pastorelli, E., Kallonen, A., & Suuronen, J. P. (2016). Methods for fibre orientation analysis of X-ray tomography images of steel fibre reinforced concrete (SFRC). Journal of materials science, 51(8), 3772-3783.

Why did you use such a long fibres and such a large amount? In literature, fibres with 30 mm are mainly used, with no more that 4wt.%. 

What we can learn from your study compared to other studies where steel fibres were used in cementitious matrix?

The deviations for compressive strains in Table 5 is too large. Why?

Author Response

Response to Reviewer 3 Comments

Point 1:

Figure 1 does not represent grading of fine aggregates. Please, provide correct grading, since you are showing that max aggregate diameter is 1 mm in Figure 1, while stating in the text to be 0.5mm.

Response 1:

Thank you for your comments.

I have deleted grading curve of fine aggregate. I don’t have grading curve.

I’m sorry for unable to provide grading curve in paper.

Point 2:

The introduction is not imporved, and backround of study is not clear. Please, provide a table with summary of properties of steel fibres (diameter, length, aspect ration, tensile strength, volume used in matrix), sand in matrix (amount, max aggregate diameter), cement (type and amount) which were used up to know in the literature. 

Response 2:

Thank you for your comments.

The introduction has been changed and background edited.

The properties of steel fibers is mention in paper on table 4.

Point 3:

Why did you use such a long fibres and such a large amount? In literature, fibres with 30 mm are mainly used, with no more that 4wt.%. 

Response 3:

Thank you for your comments.

High mixed steel fiber volume fraction were used to resistance against large deformation under blast and impact load.

Point 4:

What we can learn from your study compared to other studies where steel fibres were used in cementitious matrix?

Response 4:

Thank you for your comments.

In order to resist shock or explosion, it must be tough to resist due to the limitations of steel fiber volume fraction in conventional fiber reinforced concrete. For this reason, SIFRCC has been developed to increase steel volume fraction.

Related background has been edited.

Point 5:

The deviations for compressive strains in Table 5 is too large. Why?

Response 5:

Thank you for your comments.

As an initial step in the development of the SIFRCC, we get this result more experiments are planned and will be conducted soon.

Round 4

Reviewer 3 Report

-